# Sustainable and Resilient Urban Water Systems: The Role of Decentralization and Planning

**Nancey Green Leigh \* and Heonyeong Lee** 

Department of City and Regional Planning, Georgia Institute of Technology, Atlanta, GA 30332, USA; leehy@gatech.edu
\*   Correspondence: ngleigh@gatech.edu; Tel.: +1-770-468-8567

**Abstract:** Urban water systems face multiple challenges related to future uncertainty and pressures to provide more sustainable and resilient modes of service delivery. Transitioning away from fully centralized water systems is seen as a primary solution to addressing these urban challenges and pressures. We first review the literature on advantages, potential risks, and impediments to change associated with decentralized water system. Our review suggests that adopting decentralized solutions may advance conditions of sustainability and resilience in urban water management. We then explore the potential to incorporate decentralized water systems into broader urban land use patterns that include underserved residential neighborhoods, mixed-use developments, and industrial districts.

**Keywords:** sustainable urban water management; decentralized water infrastructure; water security and resiliency; socio-institutional impediments

## 1. Introduction

The age and centralized design of urban water management systems in the United States pose significant and increasing economic, social, and environmental costs to the communities they serve. One need only look to the recent Flint water crisis that began in 2014 to see that lead poisonings caused by contaminated urban drinking water continue to be a very real threat. In Flint alone, over 6000 children may have experienced lead poisoning. Remediating Flint's water infrastructure could cost as much as $1.5 billion [1]. Additionally, Feldscher [2] estimates the cost to society to support the affected children totals another $1 billion in social safety net spending and foregone income potential. Flint's unprecedented environmental disaster resulted from a chain of multiple lapses and failures: reliance on unproven temporary water resources, failure in corrosion control treatment, and century-long environmental injustice [1,3]. Arguably, however, the root cause of the crisis was Flint's aging infrastructure and the city's struggle to finance a water system under severe fiscal pressures [4].

While the scale and severity of Flint's water supply crisis appear especially egregious, the conditions and pressures that contributed to its crisis are not unique, rather, they are indicative of the nature of current water systems throughout the United States. As cases in point, the same year (2015) that the Flint crisis made headlines, officials also found elevated levels of lead in the water systems of Jackson, Mississippi [5], and Sebring, Ohio [6].

Most underground water pipes in the United States were installed 50 or more years ago [7]. These pipes have reached the end of their designated lifespans and require significant refurbishment. Even if these systems were still in good working order, they would require upgrades to meet contemporary environmental standards for drinking water and wastewater treatments. The cost of such modifications has outpaced service revenues. Despite the fact that water utility debt increased 33% from 2000 to 2010 [8], there remains an $11 billion annual shortfall in water infrastructure upgrade and replacement expenditures [9].

Given the costs associated with aging infrastructure, many cities struggle to provide safe water at an affordable rate. For example, while the city of Atlanta has relatively abundant water resources, it also has some of the highest water rates in the country—$325.52 per month for an average household compared to an average of $140.40 for 30 major US cities [10]. The high rates are in part attributable to upgrades mandated by federal consent decrees in 1998 and 1999 to rehabilitate and separate the city's combined sewer and water system (at a cost of over $700 million as of 2015) and to reduce sanitary sewer spills into the Chattahoochee River [11].

Concerns about cost of service are also complicated by pressure to provide more efficient water resource use and reduce system vulnerability in light of increased climate variability, heightened environmental awareness, deteriorating water infrastructure, tighter regulatory controls, changing demographics, and increasing inter-regional water conflicts [12,13]. Atlanta is also experiencing all of these pressures. A single water source, Lake Lanier, supplies over 70% of the Atlanta metropolitan area's water demand. Such heavy reliance on one water source nearly caused a failure in water supply in the city during the below-average rainfall periods from 2004 to 2008 [14,15]. The metro region has also been involved in lawsuits (referred to as the "Tri-State Water Wars") for more than two decades with Alabama and Florida over Georgia's increasing water withdrawals from their shared watershed to meet the water demands of the Atlanta metro's rapidly growing residential and jobs base.

In addition to the challenges stemming from neglected capital investments, insufficient infrastructure refurbishments, affordability concerns, and resource inefficiencies and vulnerabilities, conventional urban water systems are also inherently limited by their centralized design. These systems feature extended water collection and distribution networks and compartmentalization that result in sub-optimal outcomes [16]. Centralized design also precludes flexible system reconfigurations for changing operational conditions [17]. Compartmentalization is characterized by a lack of inter-connectedness among infrastructures that only perform limited and specialized function in the conventional urban water system. It leads to the wasting of useful resources (water, heat, nutrients) and extended network distances. The above represent the primary challenges to promoting sustainability and resiliency in urban water systems.

There have been ongoing efforts to construct alternative frameworks for urban water management as a response to the significant challenges of today's centralized urban water delivery systems. In particular, the new framework broadly referred to as Sustainable Urban Water Management (SUWM) emphasizes more adaptive and integrated management of the total water cycle and efficient water resource use through diverse and flexible multi-scalar solutions [12,18,19]. SUWM's central feature is the use of decentralized and integrated or multifunction physical water infrastructure with small ecological footprints that typically use locally available water sources, with the net effect of increasing urban water system sustainability and resiliency [12,20,21]. Water reclamation, gray water recycling, rainwater harvesting, and stormwater harvesting infrastructure are all examples of decentralized processes. A number of demonstration projects show that decentralized infrastructure does in fact result in water resource conservation, energy and cost efficiency, improved system security, and greater adaptability in configuring water systems for specific local contexts and changes in operation conditions [17,22,23]. Such infrastructure therefore directly addresses many of the weaknesses associated with centralized water systems.

Despite their benefits, the adoption of decentralized systems has failed to go beyond the demonstration phase [19,24]. Multiple studies attribute the slow adoption rate to technological entrapment stemming from socio-institutional impediments, and results in inferior technologies surviving long after that should have been replaced by new and better technologies [19,25,26]. Additional explanations include the technological uncertainty and complexity of decentralized water infrastructure [12,20].

In this article, we examine land-use scenarios that represent particularly strong opportunities to move beyond barriers to sustainable technological transition and incorporate decentralized water infrastructure. In particular, we examine the potential applications in residential neighborhoods—especially those that have historically been underserved in terms of water infrastructure capacity and maintenance—as well as in mixed-use developments and industrial areas. Adopting decentralized water systems at these smaller scales

could enable cities to test the technology and build public and institutional support that are prerequisites to longer-term, more macro-scale applications [19].

The remainder of this article is organized as follows: In Section 2, we present a conceptual framework for sustainable and resilient urban water systems. Section 3 provides an overview of the limitations of conventional urban water systems. Section 4 presents the strengths of decentralized water infrastructure as a complement to centralized water infrastructure and discusses impediments to its adoption. In Section 5, we discuss land uses and development patterns that would be particularly well-suited to decentralized water infrastructure. This article's conclusion is presented in Section 6.

## 2. Sustainable and Resilient Urban Water Systems

### 2.1. Sustainable Urban Water Systems

Before discussing the concept of resiliency, we must first introduce the broader framework of sustainability. As a normative concept, sustainability is defined as those physical and institutional practices that "meet the needs of the present without compromising the future generation to meet their own needs" [27]. Often depicted as a triangular model that balances the competing priorities of social justice, economic growth and efficiency, and environmental protection, sustainability is the most widely recognized framework used in natural resource management [28,29]. Resolving and reconciling tensions generated between different development priorities is required to achieve sustainability [30].

Figure 1 illustrates how this concept applies to sustainable urban water systems. We refer to this paradigm as "sustainable urban water management" (SUWM), to borrow the terminology of Marlow, Moglia, Cook and Beale [12]. The top of the triangle represents social justice priorities, which assert that urban water systems should distribute water resources and costs equitably and through democratic decision-making processes [31]. The bottom right point of the triangle depicts the economic goal of efficiently providing adequate water quantity and quality to support human needs and ensure the area's vitality and water security [32]. The bottom left point of the triangle depicts environmental goals, including the long-term viability and renewability of freshwater stocks and flows [32]. Conflicts between the three priority areas are inevitable. Thus, the sustainability of urban water systems requires finding a balance that responds to a range of priorities while reconciling tensions among them. This is accomplished through the cooperation of various individual and organizational stakeholders, the use of innovative infrastructure technology, and the regulation of the system by institutional actors.

A major limitation of the sustainability model is that it suggests a static, balanced system. The tensions between different priorities are continuously reshaped and cannot be resolved permanently. Internal pressures, such as growing demand for service provision and urban amenities, continually change the underlying assumptions for sustainable consensus building. Moreover, emerging social considerations may require reorganizing priorities among existing and new objectives to yield an evolving sustainability trajectory [19,33]. Additionally, urban water systems interact with multiple other external systems—including those related to climate, demographics, and the urban economy—which themselves are dynamic and can thus change the context in which the water system operates [12,13,20]. Therefore, while SUWM is a useful normative framework for urban water systems, its ability to absorb internal and external disturbances, as well as to adapt and evolve to pressure is dependent upon its design [29,34].

The goal should be to design systems that are responsive in addition to sustainable—in other words, systems that are "resilient." A resilient system can absorb a high level of disturbances, has greater capacity to reorganize itself to adapt and evolve with disturbances while maintaining essential functions, and self-directs the path of adaptation toward a more desirable state [31,35]. When a resilient system faces a major disturbance, it not only maintains the capacity to perform essential functions, it also adapts to external changes, thus maintaining the sustainability of the system [16]. This is an important distinction relative to systems that are merely "robust;" while robust systems focus on mitigating system failures through strengthening individual components, resilient systems attempt to retain and rapidly reinstate system functionalities after failure, through flexibility and diversification of functional dependencies [36,37].

Additionally, while the primary goal of a robust system is to simply persist and maintain the system's original functionality, a resilient system actually transforms itself to respond to change [29]. Given that resiliency enables a system to maintain its sustainability, we must extend the SUWM paradigm to reflect resiliency as the goal of urban water systems.

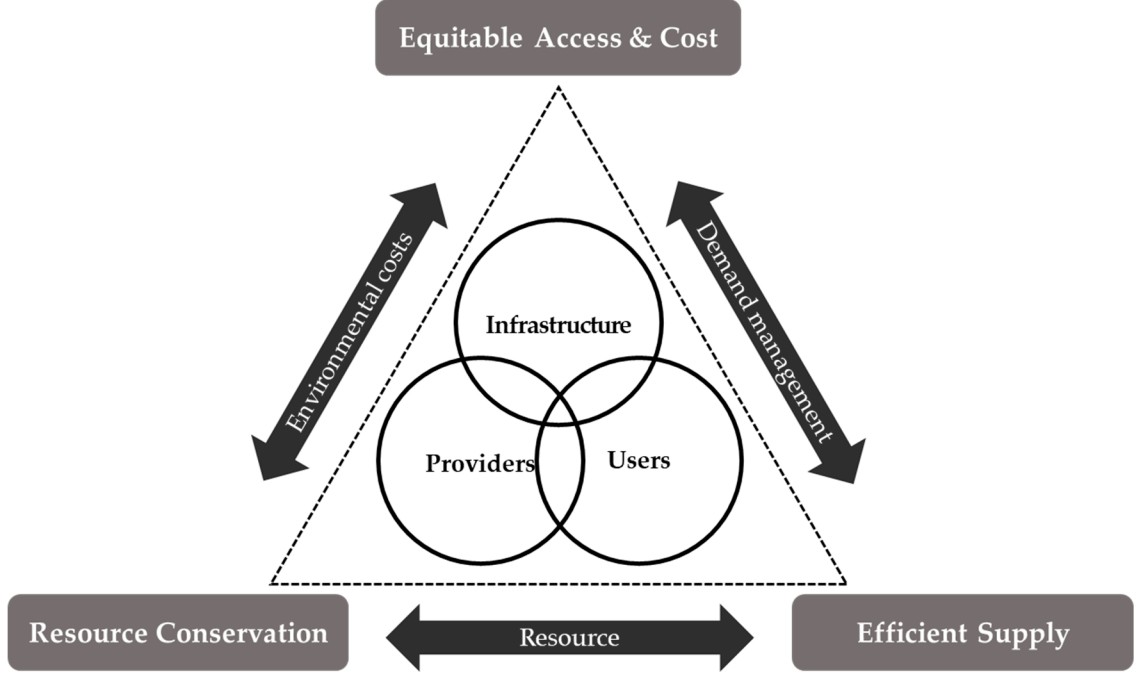

**Figure 1.** The triangular model of urban water sustainability.

Although sustainability and resiliency are highly interrelated concepts, it is important to note that objectives of resilience may conflict with that of sustainability [38,39]. That is, actions and policies that aim to achieve a resilient system at one temporal or spatial scale may negatively affect sustainability goals at another [40]. For instance, efforts to achieve a resilient urban water system are likely to increase service costs in the short term, resulting in financial instability of the city's water department or increased water prices. Effective policy and planning to achieve a sustainable and resilient urban water system recognizes temporal and spatial consequences of potential actions, as well as commonalities and dissimilarities in objectives of sustainability and resiliency in order to develop synergistic efforts [41].

### 2.2. Adaptive Management Practices for Resiliency in Sustainable Urban Water Management

Flexibility and adaptability are essential components of resiliency. Both can be achieved through the practice of "adaptive co-management." Adaptive co-management combines two emerging approaches to managing uncertainty and complexity: adaptive management, which takes a scientific approach to "learning by doing" using rapid feedback from updated scientific information; and co-management, through which system adjustments are made collaboratively [42,43]. Adaptive management is a data feedback process, while co-management is a stakeholder feedback process. Combining both feedback types enhances system responsiveness to all demands from internal and external forces. SUWM has embraced adaptive co-management and explored new urban management practices for more adaptive and integrated urban water systems [18,25,44]. In contrast to the conventional approach to urban water management employing "largely fixed and inflexible solutions," emerging adaptive co-management practices focus on diverse and flexible solutions that are at multiple scales and across technical, social, economic, and ecological spheres [18,19].

As the component processes of adaptive co-management imply, resiliency-oriented system management requires the integration of systems, agents, and institutions [36]. For example, building

agent and institutional capacity cannot be separated from ongoing discussions about building resilient infrastructure systems because system flexibility is not a predetermined characteristic of an infrastructure system. Rather, it is a discovered characteristic in a process of broader social interactions and shared learning. In addition, the adoption of new infrastructure options is not based solely on technical choices; instead, it results from strategic decisions made under a specific institutional arrangement as a response to changing conditions. Thus, building agent and institutional capacities to detect and respond to new conditions, to facilitate involvement and coordinate stakeholders, and to learn from experiences are integral to enhancing the resiliency of urban water system.

### 2.3. Resilient System Design

In addition to being integral to system management, flexibility and adaptability are also fundamental to the physical design of a resilient water system. Specific strategic considerations include assuming change and uncertainty, nurturing conditions for recovery and renewal after disturbance, combining different types of knowledge for learning, and creating opportunities for self-organization [43]. Technical approaches that incorporate these strategic considerations include designing for system connectivity, diversity, and redundancy [36].

To ensure flexible and adaptable operation, systems must be designed for both internal and external connectivity. Designs that utilize overlapping networks create internal connectivity, for example a pipe layout that have alternative service routes for water main break. Designs that connect a system to outer systems provide for external connectivity, for example, an inter-regional pipeline for water transfer. Both types of connectivity generate more options for alternative service paths and increase the ability to transfer spare capacities among water system components. Both types of connectivity generate more options for alternative service paths and increase the ability to transfer spare capacities among water system components.

There are two kinds of diversity needed in the resilient system: spatial diversity, which entails the distribution of water system assets and functions so that one adverse event does not affect all at the same time, and functional diversity, in which there are multiple means to satisfy a particular end [36,37]. There are three ways to diversify urban water infrastructure: first, drawing on a mixed portfolio of local water sources to reduce vulnerabilities associated with diminishing resources [17]; second, configuring the system so that it can run on multiple technologies as technology changes and upgrades occur over time [34]; and third, installing spatially distributed infrastructure to avoid significant degradation or failure of system function from an adverse event [36].

Lastly, designing for system redundancy enhances buffer capacities and allows for multiple service paths in order to respond to unexpected increases in water demands or operational failures [36]. Urban water infrastructures can achieve redundancy in multiple ways: by increasing the capacity of existing infrastructure, by adding preventive features in anticipation of system failures, by utilizing modular design so that components can be moved or reutilized [34], and by installing intermediate components like graywater systems that can bypass certain parts of the urban water system to increase resource efficiency.

The suggested physical features—connectivity, diversity, and redundancy—interdependently contribute to improving system resiliency. Thus, their synergistic relationship must be considered in designing a resilient urban water system. System design that focuses on a single physical feature can limit the system's effectiveness to a specific event and lead to negative resiliency consequences. For example, increasing reserve capacity of a water-supply system may allow for reliable service delivery in the event of a temporal breakdown. Alternatively, if the system is subject to a substance intrusion, then a large reserve capacity would prolong the recovery time [45].

### 2.4. Model of the Resilient Urban Water System

Given the literature on resiliency discussed above, a conceptual framework for resilient urban water systems is depicted in Figure 2. Complexities and uncertainties associated with unpredictable shocks and

slow-burn stresses from interdependent systems and internal pressures are embedded in urban water systems and threaten their long-term sustainability. To tackle such challenges, the model shows that urban water systems must be responsive, such that they maintain capacity for recovery after disturbances and create self-organizing capacities so that the system can automatically evolve in response to internal and external disturbances [43]. Overall, the model illustrates how the building blocks of these systems are flexible and adaptable urban water infrastructure systems (whose physical designs incorporate network connectivity, diversity, and redundancy), institutional capacities, and resilience strategies.

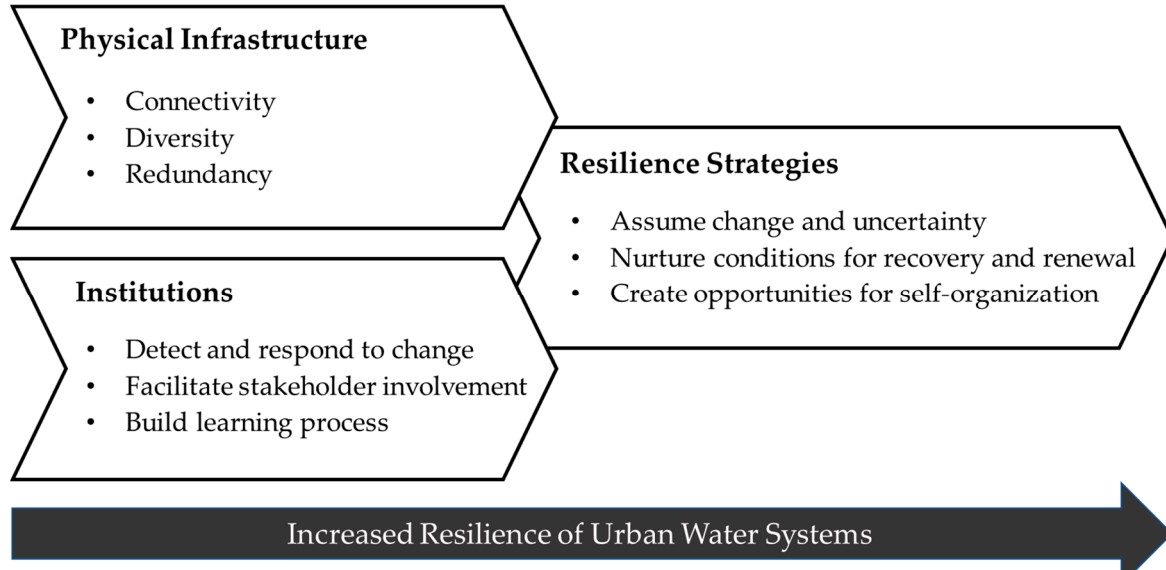

**Figure 2.** Conceptual framework for resilient urban water systems.

## 3. Limitations of Conventional Urban Water Systems

### 3.1. System Inefficiencies that Undermine Sustainability

Centralized urban water systems provide services at relatively inexpensive costs to meet human needs. They are especially effective in urban areas where population and economic activities are concentrated. The historic development of centralized water system design enabled professional operation and maintenance using increasingly advanced technologies to meet water quality standards. However, there are several sources of inefficiency that undermine the sustainability of current urban water systems, namely, inefficiencies regarding design, cost, energy, natural resources, and management. Regarding design, every centralized urban water system requires a long-distance water pipe network, as well as specialized facilities for different treatment processes [16,21]. The fact that each facility is designed for a singular purpose results in wasted opportunities for more efficient and ecological urban water management. For example, despite a growing recognition of stormwater reuses, current stormwater drainage systems designed for flood protection consider collected stormwater only in relation to drainage, while overlooking stormwater's potential as an alternative water source [46].

As a consequence of the amount of pipeline required, centralized urban water systems have significant costs and inefficiencies. Installing pipe networks typically makes up the largest share of water investment. In the United States, the transmission and distribution system constitutes 45.9% of total national water investment [47]. Approximately 23 million cubic meters of potable water is lost annually in the process of transporting the water from treatment facilities to users [7]. The costs of managing, operating, and replacing large, centralized systems increase as systems age. Communities report 240,000 cases of broken water pipes annually, and deteriorating, 19th-century pipe networks are often to blame [47].

Centralized urban water system design also makes inefficient use of energy. Water-carriage sewage systems are similarly wasteful of useful resources, including energy, water, and nutrients [12]. Pabi et al. [48] estimate that 1.8% of total United States electricity use (69.2 billion kWh) goes to providing public drinking water and operating public wastewater treatment systems. The amount of electricity used for water and wastewater systems is expected to grow to meet increasing water demand and tighter regulatory controls [49].

Centralized urban water systems also waste environmental resources, which can have severe ecological impacts [32]. Centralized urban water systems require large-scale water collection and treatment facilities, which dramatically alter natural hydrology systems and often lead to unexpected environmental consequences, such as stream depletion, shoreline erosion, and other negative biological outcomes. The unexpected consequences and costs are not limited to natural systems. Ultimately, these ecological impacts have social and economic costs for humans [32]. As a case in point, large-scale modifications to natural water systems can increase stormwater runoff and, consequently, vulnerability to floods [50]. Contamination of water resources and changes in local climate can also have negative health impacts and reduce productivity in agricultural and fishery sectors [24].

Conventional water systems also suffer from inefficient management. While their centralized design would imply a centralized management structure, in practice the management of these systems is fragmented [16]. For example, the entity that manages the wastewater treatment may not manage the drinking water system. The fragmented management structure impedes the institutional capacity to incorporate newer systems that bridge multiple processes, such as graywater treatment.

## 3.2. Barriers to Resiliency

The lack of flexibility and adaptability in conventional water systems prevents them from being resilient. First, high fixed costs impede the ability to retrofit or upgrade these systems. Fixed costs make up 70% to 80% of total investment in the system [17]. Such costs also motivate communities to utilize these systems for as long as possible. Centralized systems are typically designed for a useful life of up to 100 years; their design is meant to endure rather than accommodate potential change [17]. Consequently, these systems are highly inflexible, and their reconfiguration possibilities allow for small cost-savings potential. Additionally, due to their reliance on a limited number of water sources and the hierarchical network structure, centralized water systems are more vulnerable to changing patterns of precipitation and sudden or unexpected climate events such as prolonged floods or droughts [19,51].

## 4. Decentralized Solutions for Sustainable Urban Water Management: Advantages and Barriers

Decentralized water infrastructure typically refers to small and medium-sized water infrastructure that uses locally available water sources including gray water and stormwater run-off, and work independently or combined with conventional water infrastructure [20,21]. Wastewater recycling, gray water, rainwater and storm harvesting infrastructure are generally considered to be decentralized water infrastructure, Low Impact Development (LID) techniques such as bio-retention facilities and permeable pavers can also be included. While sizes and scales matter, these are not the sole features that distinguish decentralized water infrastructures from conventional infrastructures. Rather, the important difference lies in the integrative functionality of decentralized infrastructures working across urban water management sectors which are traditionally compartmentalized. This feature enables the diversification of water supply options and extended internal water circulation within urban water systems through the pathways. The top half of Figure 3 represents a linear water flow achieved in the conventional centralized water system often described as the take, make, waste approach [23]. The decentralized water system that uses alternative water sources is depicted in the bottom half of Figure 3, which illustrates the circulation of water pathways within the urban water system.

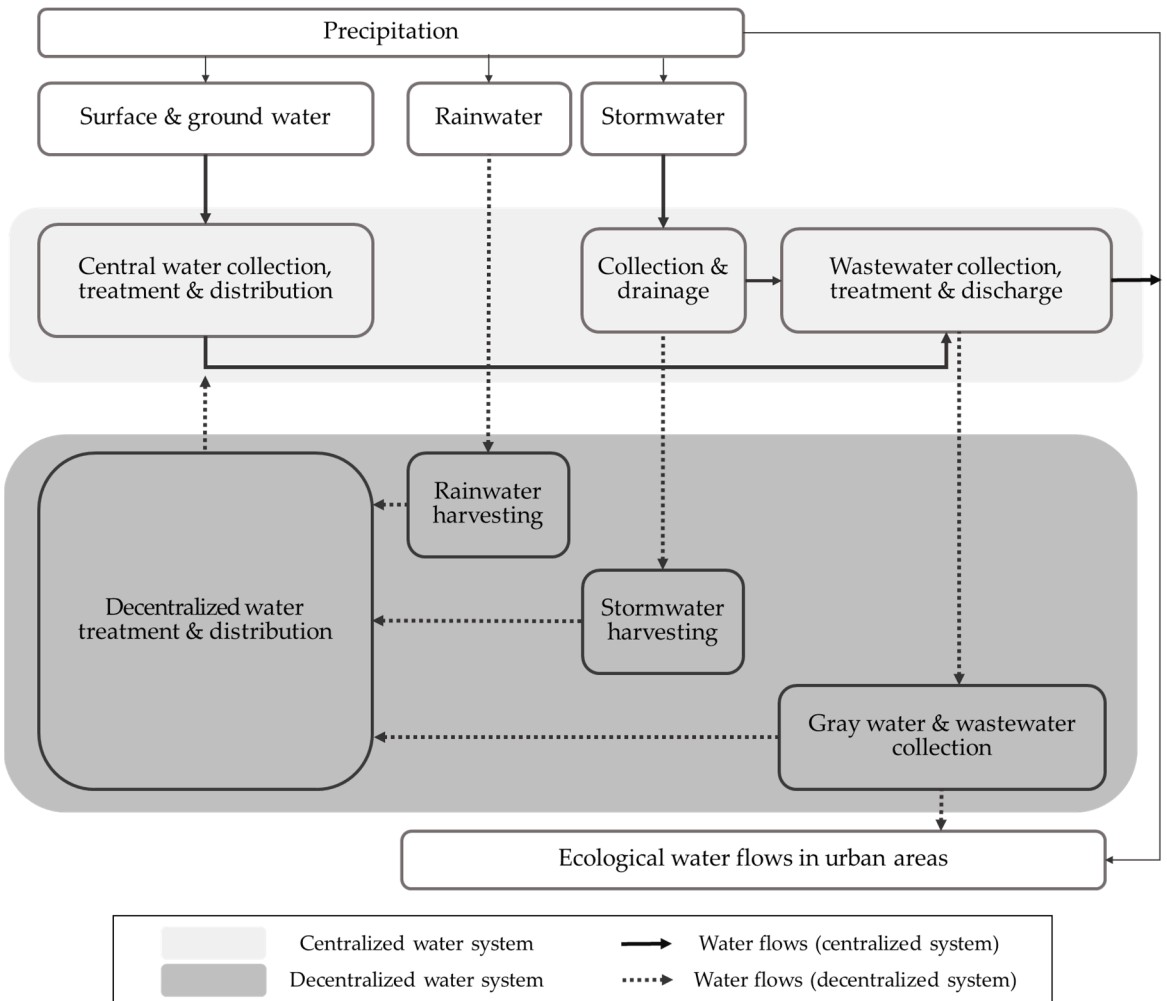

**Figure 3.** Centralized and decentralized components and pathways in urban water management.

## 4.1. Advantages of Decentralized Infrastructure

The perceived shortcomings of conventional water systems prompted substantial interest in more sustainable approaches to urban water delivery systems, which in turn led to the development of the SUWM paradigm. SUWM emphasizes the decentralization of urban water infrastructure as a fundamental physical requirement. The advantages of decentralized urban water systems address all of the major limitations of conventional centralized systems today, in that decentralized systems are integrated and sustainable, as well as resilient. Moreover, decentralized urban water technologies can complement existing centralized urban water systems. As additions or partial improvements to the original system, decentralized technologies introduce sustainability and resiliency capacity into the water system without requiring the wholesale replacement of existing infrastructure.

By increasing system interconnections, decentralized water systems are more integrated than centralized systems. Indeed, the key feature of urban water infrastructure under the SUWM paradigm is integrated functionality across urban water management sectors that are traditionally compartmentalized. While centralized systems operate on the premise of "water in, water out" (i.e., processing drinking water and wastewater sequentially), decentralized water systems utilize multiple water sources, feature greater path diversity, and extend internal water circulation, thus enabling more efficient resource usage overall. This integration underlies the sustainability and resiliency of decentralized water technologies.

Decentralized water systems' efficient use of resources is a key reason they are more sustainable than conventional urban water systems [12,20,22]. In particular, decentralized water infrastructure utilizes non-conventional water sources and fit-for-purpose water supplies. This is a response to

the mismatch between the production and utilization of potable versus non-potable water through a centralized system. Only a small portion of potable water supplied through conventional water systems is actually used for potable purposes [12], while approximately 40% of wastewater generated from single households can be reused as gray water for non-potable purposes [52]. Stormwater is another potential source of usable water that is not leveraged in existing centralized systems but that can be incorporated into decentralized systems. Theoretically, the rainfall collected from one square kilometer of land in a city like Atlanta (where average precipitation totals 49.6 inches per year) can produce approximately 333 million gallons of water annually, which can support the annual water demands of 23,000 people, assuming they each use 50 gallons of water per day [46]. Utilizing such alternative water sources in urban catchment areas, decentralized technology can contribute to reducing water demands from conventional water systems by 30% to 60% [22,53]. Reducing the amount of water collected for centralized water systems increases environmental flows that are critical for restoring and maintaining the health of an ecosystem.

The water saving potential of decentralized technology also enhances the cost efficiency of water systems by reducing their energy consumption and minimizing their ecological footprint, both of which benefits also improve the sustainability of the water system overall [23]. Cost–benefit analyses for decentralized technologies such as wastewater treatment, gray water, and rainwater harvesting have demonstrated positive economic by reducing energy demands for water treatments and transfer. Xue et al. [54] found that on-site gray water treatment decreased system energy consumption by more than 50%. Such savings can be augmented by combining multiple decentralized technologies. For example, a water system with on-site gray water treatment and rainwater harvesting is estimated to consume about 25% of the total energy of a conventional water system [54]. The cost-saving potential of decentralized technologies is even greater when one takes into account the ability of decentralized systems to mitigate peak water demand and thus reduce the need for capital investments to increase the capacity of existing treatment facilities [55]. Decentralized systems also help minimize the ecological footprint of urban water systems through water resource recovery, which can also be extended to include nutrient recovery [20,23,56], and utilization of sludge and water to generate electricity to power the system [57].

Decentralized water systems also have the potential to enhance equity and environmental justice within the communities they serve. For example, consumers can benefit from the cost efficiencies outlined above if they translate into lower overall service prices. Additionally, because decentralized technology can be designed to supplement existing water systems, decentralized solutions can be implemented in targeted areas to correct historical imbalances in infrastructure investment and improve water service and water quality to specific underserved constituencies.

Combining decentralized water infrastructure with a centralized system can also increase the resiliency of that urban water system by reducing the vulnerability of the system to both shocks and gradual change. By design, decentralized urban water systems have greater capacity to cope and adapt: they can draw on a diversified portfolio of water sources, increase system buffer capacity by reducing potable water demands, and utilize multi-scale networks and pathways [17,20]. Decentralized system components that are geographically dispersed and independently operated also provide safe-failure features by limiting impacts of system failures to smaller geographic areas and preventing a domino effect of failure among other system components [58]. Furthermore, due to lower capital intensity (because of lower fixed costs) and shorter construction timelines, communities can deploy decentralized infrastructure more rapidly to respond to external disturbances, such as climate change and demographic variability, and with less operational risk [19,59].

Decentralized infrastructure is not just adaptable, but also flexible. It can be low-tech, low-cost, and flexible in its service boundary [22], enabling the infrastructure to respond to specific local institutional requirements and demand conditions [20]. Decentralized water infrastructure can be adopted and managed at multiple scales, from the individual homeowner to the region [59]. Thus, it can be a tool for local communities with to manage urban water problems and help to address

localized concerns over water through innovative approaches based on synergies between local actors and local conditions. For example, local farmers in the town of Maldon (the Victorian Goldfields in Australia)—who previously suffered from limited water resources—gained access to additional local water sources through the installation of a water transport pipe linked to local mine operators—who previously charged for the pumping and containment of underground water in mine operations [22,60]. Officials can also leverage the multi-scale nature of decentralized urban water systems and the corresponding expansion of water access to increase community engagement in water planning and enhance resident awareness of water-related issues.

The above advantages make decentralized infrastructures particularly compelling for addressing the problems of growing cities that otherwise must make large investments to expand existing centralized water facilities, secure more water resources, and remediate ecological impacts due to excessive water withdrawal [17]. At the same time, these characteristics also respond to the needs of declining cities, by optimizing system scale to reduce operational costs for underutilized water facilities [17]. The advantages also address environmental and social vulnerabilities particular to lower-income communities; therefore, decentralized water systems offer many potential benefits for society at large and for communities that are underserved by current infrastructure.

### 4.2. Impediments to Adoption

Despite the potential benefits of decentralized water systems, there are multiple barriers to their adoption. Most significantly, path dependencies related to both infrastructure technology and management structures favor existing systems over new ones. Because of the large fixed costs of existing centralized water systems, agencies often make maintenance and upgrade decisions based on those that will extend the useful life of the system [61]. This approach favors tweaks to the existing system, rather than introducing new technology that reduces system costs in the long run [62]. Although the original investments in centralized water systems are sunk costs, they create a "lock-in" effect for the original and "proven" technology, which only further biases investment decisions toward the status quo and makes it harder to adopt decentralized technology [61,63].

Technological path dependency is not only a result of the perceived economic value of new technology, but also of the existing structure of actors and institutions that manage the system [25]. The compartmentalization of the water sector across water supply, sewage, and stormwater functions has been embedded into the division of managerial responsibility for water service provision, operation, and maintenance [16]. This fragmented administrative structure does not lend itself well to the management of an integrated system; were such infrastructure to be built, reorganization of roles and responsibilities would likely be necessary. Institutional actors may also be wary of increased task burdens or reporting standards, particularly given that the unproven nature of urban water system technology inherently comes with public health risks that require monitoring and potential intervention [33]. For these reasons, there is a lack of institutional will to adopt decentralized water system technology [24,26]. The lack of legislation, public acceptance, and community involvement in planning further weaken the political support for such change [64].

Institutional decision-making criteria also do not consider the full range of costs and benefits of centralized and decentralized infrastructure. In addition to making investment decisions premised on the "sunk costs" discussed above, conventional cost-benefit analyses underestimate the social and ecological costs of centralized water systems [24,65]. At the same time, decision makers may overlook many social and ecological benefits of decentralized water systems, such as increasing community water security and conserving ecological water flows, as these impacts are broad in scope and more difficult to quantify [66]. A conventional cost-benefit analysis could even construe the water saving potential of decentralized technologies as a threat to the financial stability of the water department on the premise that decentralized systems would reduce service revenues. Additionally, the time horizon of the cost-benefit analysis is crucial. A short-term focus can indeed undermine the perceived

cost-efficiency of decentralized water infrastructure, as the cost of the new infrastructure must be amortized over its useful life [12].

Another perhaps counter-intuitive barrier to adoption of decentralized water infrastructure is the flexibility of the technology. In many respects, this flexibility is a relative benefit of decentralized water systems because it allows for custom, context-sensitive solutions; however, the flexibility also makes system design and management more complex. Decentralized urban water systems consist of overlapping facilities for water collection, storage, and distributions that occur over multiple spatial scales and duplicate water networks for potable and non-potable water [20,67,68]. Engineers can also design decentralized components to link up with the existing centralized system so that the technology can complement the existing system, rather than require a total system overhaul. The result is the interconnection of diverse technologies and new patterns of interaction among even existing system components. Such complexity impedes the establishment of best practices that ensure successful application of the technology and promote its increased adoption and diffusion.

The effectiveness of decentralized systems is highly dependent on system configuration and specific contextual factors, including the system's selected water sources, network scale, topology, and external subsystems [12,69]. The untested interactions among system components can cause unintended long-term negative consequences that require action to ameliorate. For example, gray water systems can increase the concentration of pollutants in wastewater, which can lead to corrosion and soil deposits in the sewer pipes, which yield increased wastewater treatments costs [20]. The tradeoff of flexibility is therefore increased complexity in implementation and management decisions, but with an appropriately broad and long-term consideration of the overall costs and benefits, the value proposition of decentralized water systems remains promising. See Table 1 for a brief summary of advantages of and impediments to decentralized water systems.

**Table 1.** Summary table for advantages of and impediments to decentralized water system.

| | |
|---|---|
| **Advantages** | Reduced water withdrawal/increased environmental water flow |
| | Reduced energy requirement |
| | Ability to improve water service equity |
| | Water sources diversification |
| | Increased buffer capacity |
| | Safe-failure feature |
| | Lower capital intensity & shorter construction timeline |
| | Adaptability to local contexts |
| **Impediments** | Attachment to proven technologies |
| | Fragmented water administrative structure |
| | A lack of legislation, public acceptance, and community involvement |
| | Concerns over cost-efficiency and financial stability |
| | Complexity in system adoption and management |

## 5. High-Opportunity Areas for Decentralized Water Systems

When water system solutions are integrated into broader thinking and decision-making processes about land use, the incorporation of decentralized technologies can naturally follow; however, test applications of decentralized water systems will be necessary to prove the efficacy of the technology. We identify three land use scenarios that represent particularly strong test applications for decentralized water infrastructure: mixed-use development, industrial areas, and residential neighborhoods—especially those that have historically been underserved in terms of water infrastructure capacity and maintenance.

### 5.1. Underserved Residential Neighborhoods

Safe drinking water is crucial to maintaining health and safety, yet unequal access to urban water services has been well documented [70]. Low-income and minority communities are more likely to be exposed to unsafe drinking water due to disproportional infrastructure conditions and noncompliance

with federal standards for water systems [71,72]. Furthermore, disadvantages groups experience greater health and social vulnerabilities from intermittent events such as lead poisoning, flooding, and sewage overflows and disasters like hurricanes because their housing is more likely to be located in neighborhoods with aging infrastructure and greater hazards exposure [73].

Low-income households also face a greater water cost burden as a share of their household income. According to a Circle of Blue [10] study that examined water prices in 30 major cities in the United States, the average family of four pays $1685 annually for water and sewer, representing 7% of household income for a family at the poverty line. Since system investment needs for repair or replacement of existing pipelines, as well as adding additional water treatment capacity increased 106% between 1990 to 2006, household water bills are expected to continue to grow [70].

Decentralized water systems can be instrumental in correcting neighborhood service disparities and decreasing resident water cost burdens by reducing both water cost and usage. Some cities already have subsidies or programs in place to address water costs for low-income residents. For example, Atlanta's Care and Conserve program provides conservation retrofits and plumbing repairs who meet low-income eligibility requirements [74]. Decentralized water systems can augment these resources and provide deeper, long-term cost savings to more residents by tackling the problem closer to its source: by replacing or fixing the actual water infrastructure.

### 5.2. Mixed-Use Development

Mixed-use development—which includes New Urbanist development, suburban town center development, and transit-oriented development—is another natural fit for decentralized water technology because of its sustainability relative to sprawling greenfield development. As the term suggests, this type of development integrates multiple land uses within the same project. The uses can be vertically integrated, in which different uses are "stacked" on top of one another, as well as horizontally integrated, in which different uses are placed side by side. The result is a more compact collection of diverse uses than that seen in more unplanned or sprawling environments. The compact, mixed-use design makes such development more sustainable by enabling reduced vehicle use, more efficient provision of public services, and a smaller ecological footprint [75–77].

From an engineering perspective, the design also makes mixed-use development well-suited to testing decentralized water systems. For example, such development requires shorter water networks, which reduces installation costs and improves operation efficiency by lessening energy and water losses in water transportation [57]. The shorter network also offers the opportunity to adopt sewage heat recovery techniques as part of the system architecture. These techniques provide additional energy savings potential in the range of 17% to 58% for a typical residential house [78]. Mixed-use development can also enhance the efficiency of gray water treatment processes, as the surplus gray water generated by residential units can be used to satisfy the gray water supply deficit among commercial and industrial properties. One study found that the net present value of a shared gray water system like this was 2.3 times higher than that of an individual gray water system [54]. Lastly, considering its lower capital intensity, decentralized water systems can be used as a tool for coping with the future uncertainty in water demand associated with new urban development.

Because mixed-use development is becoming more popular, it represents a growth opportunity for the application of decentralized water systems. Additionally, because mixed-use development is a type of new construction or ground-up redevelopment—and often includes new streets, in addition to new buildings—it offers the opportunity to incorporate decentralized water systems from the earliest phases of a place's planning.

### 5.3. Urban Industrial Development

The industrial sector is the largest consumer of non-consumptive water uses, and its water demands can be satisfied by lower-quality water [79]. Many urban industrial actors also face pressure to use water more efficiently due to increasing water prices and limited water availability in urban areas.

Regenerative water technologies like those incorporated into decentralized water systems directly align with these users' demands by allowing industrial facilities to use lower-quality water and realize both cost savings and environmental benefits [80]. Stormwater harvesting and gray water technology are particularly well suited to supplying non-potable water at reduced costs. Also, these technologies can lower risks associated with a long-term and a temporal shortage of industrial water supply by providing alternative water sources. In light of the importance of water prices and availability in industrial recruiting and retention, decentralized water systems can be a sustainable and enduring source of competitive advantage.

As with mixed-use development, industrial development is a high-growth sector of real estate development. Driven in large part by the rise of e-commerce and the demand for new facilities to meet ever-shortening product turnaround times, industrial construction is powered by strong structural supply and demand fundamentals. As such, the integration of decentralized water systems into new industrial development provides a potential means of scaling the technology's adaptation.

Industrial users could also increase the amount of usable non-potable water—and thus water cost savings—available to them by partnering with residents to adopt a shared a decentralized water system that serves both groups. For example, Salisbury, Australia, retained a wool processing company that was considering relocation due to the high costs of fresh water and sewer disposal by supplying cheaper, non-potable water collected through stormwater harvesting facilities and wetlands [22,81]. The company and local government formed a joint venture to undertake the project. The project benefits extended beyond the company to residents, as the project provided over 3500 Salisbury residents access to non-potable water for outdoor watering purposes, such as gardening. Furthermore, the stormwater treatment features were designed to be local amenities and green spaces that the community uses for recreational and educational purposes [22].

## 6. Conclusions

The development of the SUWM paradigm and innovative water technologies offers ways to rethink the current organization and operation of urban water systems. It also shows how urban water systems, can be designed for greater sustainability and resiliency. Clearly, the innovative integration of decentralized water technologies in urban water infrastructures offers the potential for urban water systems to be more efficient, ecological, robust, and adaptable. Multiple institutional and technological impediments hinder such a technological transition, but these could be mitigated by opportunities to test decentralized water systems at a small scale. With more experience and concrete examples to point to, decentralized water technology can be optimized, and institutional leadership cultivated. Such testing could also build the political support and public acceptance needed to invest in decentralized urban water system infrastructure on a broader scale.

This study identified three urban land use scenarios that particularly stand to realize the benefits of decentralized water infrastructure—namely, residential neighborhoods, mixed-use developments, and industrial zones. These applications can demonstrate the potential of decentralized water technologies as a tool for responding to broader urban issues and concerns. In these contexts, specifically, decentralized water technologies can alleviate disproportionate access to urban water services, enhance the energy efficiency of compact and mixed-use development, and secure preferable business conditions for urban industries. Additionally, incorporating decentralized solutions into these contexts can produce substantial synergies for both the adoption of this technology and responsiveness to other issues that urban communities face. Linking the potential of decentralized water infrastructures with the broader urban development agenda can attract more financial resources to the implementation of decentralized systems, increase efficiency and performance of the larger urban built environment, and create opportunities for community involvement and inter-sectoral collaborations.

This review details the resiliency potential of decentralized water systems in their own right and posits that these solutions have broader potential to advance urban social and economic development goals. Future research is needed on the interconnections and synergies between these technologies and

other urban socioeconomic issues and land use patterns. Subsequent studies of decentralized water systems that are contextualized within their potential applications to urban development issues are warranted to help fulfill the promise of this technology.

**Author Contributions:** N.G.L. has performed the conceptualization, investigation, and writing the original draft. H.L. has contributed to detailed investigation, visualization, and editing the writing.

**Funding:** This research is funded by NSF Grant #1441208 (RIPS Type2: Participatory Modeling of Complex Urban Infrastructure Systems (Modeling Urban Systems)).

**Conflicts of Interest:** The authors declare no conflict of interest. The funders had no role in the design of the study, in the collection, analyses, or interpretation of data in the writing of the manuscript, or in the decision to publish the results.

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
