# Peer review of "Sustainable and Resilient Urban Water Systems: The Role of Decentralization and Planning"

_sustainability, doi:10.3390/su11030918_

Reviewer 1 Report

The goal of the paper is defined as 1) literature review on decentralized water system designs. 2) how to incorporate decentralized systems into existing urban water infrastructure.

Although I agree with authors who emphasize the advantages of decentralized systems, it seems that the authors solely focus on the disadvantages of centralized networks in the introduction. While it is stated in the abstract that they would also mention the advantages of the centralized system. Centralized water supply network was an effective solution in the early 90s as it provided a reliable water which meets the water quality and fire flow regulations.

As authors focus on examples from the USA, I find it beneficial to include it also in the title. Similarly, the sections after introduction focus on water-related issues specific to the cases in the USA. As an alternative, authors can include examples from Europe etc. where the centralized networks have been used and aging as well. Note that, the problems in Europe would differ. For example, the problem is that centralized systems are not cost effective due to the decreasing rural population.

Authors should consider the following paper which investigates the centralized to decentralized transition scenarios in engineering and sustainable planning point of view, as it is missing from the current version of their research. How engineering and design point of view, this transition would be possible? Specifically, later on, the paper is not providing physical design examples.

Aydin, N. Y., Mays, L., and Schmitt, T. (2014). "Technical and Environmental Sustainability Assessment of Water Distribution Systems."https://link.springer.com/article/10.1007/s11269-014-0768-y

Aydin, N. Y., Mays, L., and Schmitt, T. (2014). "Sustainability Assessment of Urban Water Distribution Systems." https://link.springer.com/article/10.1007/s11269-014-0757-1

I think the paper is very well written. However, there is a gap between the introduction section and section 2. Sustainable and resilient urban water system. In the introduction part, there is so little reference to sustainability and resilience. As a reader who already knows that these concepts are interrelated, I can understand why sustainability and resilience are mentioned in the following sections. However, a reader who does not have this knowledge would be confused why authors jump from advantages and disadvantages of centralized water systems to sustainability and resilience concepts.

The major problem in section 2.1 is the assumption that resilience enables a system to be sustainable or to maintain its sustainability.  I suggest authors the following papers that mention the difference and similarities. Although some systems can be resilient and sustainable at the same time, it cannot be generalized as a fact. One example I provide here is from the below reference:

In the community development context, sustainability initiatives tend to focus on preserving traditional methods of resource use, livelihoods, environmental knowledge, and environmental resources. In contrast, resilience initiatives tend to focus on adapting to new conditions, creating innovative uses of traditional knowledge, creating new environmental knowledge, and improving living conditions and employment (Lew et al., 2016) Lew, A.A., Ng, P.T., Ni, C. (Nickel),Wu, T. (Emily), 2016. Community sustainability and resilience: similarities, differences, and indicators. Tour. Geogr. 18:18–27. https://doi.org/ 10.1080/14616688.2015.1122664.

Marchese, D., Reynolds, E., Bates, M. E., Morgan, H., Clark, S. S., & Linkov, I. (2018). Resilience and sustainability: Similarities and differences in environmental management applications. Sci Total Environ, 613-614, 1275-1283. doi:10.1016/j.scitotenv.2017.09.086

The paper is very well written. A short note; I found some sentences too wordy. one E.g. Line 169-173. Maybe authors should consider creating tables for some cases, that would help to summarize some concepts and avoid repetitions.

Technological capacity/Connectivity/internal connectivity and external connectivity conceptually very vaguely defined. Also -line 186-190: I am not sure what authors mean by multi-scale networks.

Lines 180-204: Often times increasing the capacity in water systems indicated to provide resilience while increasing the capacity not necessarily resilient or sustainable. Please see the conclusion part of the following paper.

“Increasing capacity (e.g. by increasing tank size) may not always improve resilience, and may even prolong the system's recovery time significantly following removal of the stress. It is therefore necessary to assess both the properties and performance of any intervention designed to improve resilience.”

Diao, K., Sweetapple, C., Farmani, R., Fu, G., Ward, S., & Butler, D. (2016). Global resilience analysis of water distribution systems. Water Res, 106, 383-393. doi:10.1016/j.watres.2016.10.011

In addition, connectivity and redundancy that author provided as technical capacity dimensions focus on mainly capacity building. In this case, it would be useful to be more explicit about “capacity” and to give clear definitions for both. Even if authors intended to make it explicit, as a reader I cannot follow it. Otherwise, capacity as a metric seems redundant.

Lines 232-234: Do authors have any information that they can provide for a cost comparison of replacing an aging infrastructure and decentralizing the system?

Line 240: how can decentralized systems cope with the tighter regulatory controls?

Line 296: It is not clear from Figure 3 how connections and flows of the following terms “feature greater path diversity, and extend internal water circulation” illustrated? Perhaps a definition for these would be useful. Because greater path diversity can be provided in centralized as well as decentralized systems. In fact, the water distribution systems are designed with a certain level of redundancy to provide path diversity. 

A table summary would be great for the sections that mention the disadvantage/advantage of both systems. Specifically, Section 4.1. is too long and some concepts are repeated here and throughout the paper.

Authors mentioned earlier the dimension of uncertainty for resilience and sustainability, yet the uncertainty is not a part of their argument in incorporating decentralized systems into existing urban water infrastructure (Section 5. High-Opportunity Areas for Decentralized Water Systems).

Reviewer 2 Report

A paper presents advantages, disadvantages,  risks, and impediments to change associated with decentralized water system design. Sustainable urban water management was presented. Authors suggest that this model is  a static, balanced system and this is major limitation its. Important part of this study is disscuss about decentralized solutions for sustainable storm water management where advantages (integrated and sustainable, as well as resilient, complementing existing centralized urban water systems, reducing energy consumption and minimizing their ecological footprint in water systems) and barriers like for example lack of legislation, public acceptance, and community involvement  in planning this systems. Generally presented study is interesting, is review state of knowlenge and problems with water management in urban areas and giving solutions for futher to sustainable  water management. The paper is accurate for Sustainability journal and should be interesting for readers.  I have only one comment to sentence in page 6, line: 222-226: It is true but now is better awarenses of users and in many countries are systems for reuses of stormwater. You can find some informations in following papers: Managing urban stormwater: harvesting and reuse. 2006. Department of Environment and Conservation NSW;Hatt, B. E., A. Deletic, and T. D. Fletcher. 2006. Integrated Treatment and Recycling of Stormwater: A Review of Australian Practice. Journal of Environmental Management 79, no. 1: 102–113; Philp, M., McMahon, J., Heyenga, S., Marinoni, O., Jenkins, G., Maheepala, S. and Greenway, M. (2008). Review of Stormwater Harvesting Practices. Urban Water Security Research Alliance Technical Report No. 9

Reviewer 3 Report

The review work concerns sustainable and resilient urban water systems: the role of decentralization and planning,  in which the choose of the analyzed works was made selectively. Also, quite old works from 2005, 2009, etc. were taken into account. The literature review needs to be more critical. Please compare the results in this study with those in previous studies. Weak conclusions. There is  lack of careful analysis of the results of many experiences of cited papers. Articles do not indicate a detailed discrepancies in published research results, so lack of synthesis of conclusions resulting from the survey review occurs. Presented figures have bad quality, small font, etc. (it should be the same as the text) All the Figures should be improved. Abstract should be more concise.
